# Hammer Throw: a Pilot Study for a Novel Digital-Route for Diagnosing and Improving Its Throw Quality

**Bingjun Wan** [1,†], **Yuanyuan Gao** [1,†], **Ye Wang** [2], **Xiang Zhang** [3], **Hua Li** [2] **and Gongbing Shan** [1,3,4,*]

1. School of Physical Education, Shaanxi Normal University, Xi'an 710119, China; bingjunw55@snnu.edu.cn (B.W.); gaoyywf@snnu.edu.cn (Y.G.)
2. Department of Mathematics & Computer Science, University of Lethbridge, Lethbridge, AB T1K 3M4, Canada; ye.wang3@uleth.ca (Y.W.); hua.li@uleth.ca (H.L.)
3. Department of Physical Education, Xinzhou Teachers' University, Shanxi 034000, China; xiangzhang@xztc.edu.cn
4. Biomechanics Lab, Faculty of Arts & Science, University of Lethbridge, Lethbridge, AB T1K 3M4, Canada
* Correspondence: g.shan@uleth.ca; Tel.: +1-403-329-2683
† The authors contributed equally to this work.

**Abstract:** The world record of the hammer throw has not been broken since 1986. This stagnation is multifactorial. One dominant factor could be the lack of evidence-based scientific/biofeedback training. This study aims to identify key parameters influencing throw quality and structure a new digital method for biofeedback training. Wire-tension measurement and 3D motion capture technology (VICON 12-camera system) were applied in quantifying and comparing throws of a national-level and a college-level athlete. Our results reveal that multi-joint coordination influences heavily on wire-tension generation. Four phases, i.e., initiation, transition, turns, and throw, play various roles in evaluating the quality of a throw. Among them, the transition, the third turn, and the throw display explosive/rapid increases of tension. For improving the effectiveness of the skill, the whip-like control and proper SSC (stretch-shortening cycle) of muscle groups involved should be established through years of training. Furthermore, our study unveils that quick and complex full-body control could be quantified and characterized by four key parameters: wire-tension, hand- and hip-height, and trunk tilt. Hence, a wearable digital device with tension and three Inertial Measurement Unit (IMU) sensors would have great potential in realizing real-time biomechanical feedback training in practice for evaluating and improving the efficiency of various training programs.

**Keywords:** 3D motion capture; wearables; phase definition; tension; multi-joint coordination; whip-like control; SSC; biomechanical feedback training

## 1. Introduction

The hammer throw is one of the most common track and field competitions, but compared to other events, it has the longest standing world record (86.74 m), established by Yuriy Syedikh from former USSR in 1986 [1]. This more-than-three-decade stagnation is multifactorial. There is no doubt that the hammer throw owes itself to complex human motor skills, involving quick body spins, dynamic balancing, and explosive power generation. Due to the speed of the sport and the invisibility of forces applied, coaches have only been able to guess "what works" for their training. Therefore, one dominant factor for the stagnation could be the lack of evidence-based

scientific/biofeedback training [2–4]. While extensive 3D motion analyses technologies do exist, practitioners find that they are too cumbersome to be useful for training [2].

Apart from the impracticality of the current motion capture technologies in training practice, there is a gap between kinematic and kinetic data of the throw control. Previous kinematic studies on the hammer throw mainly focused on some easy-to-measure parameters. For example, Dapena [5] argued that the hammer head velocity was the most important contributing factor to the distance of the throw, but he was unable to elaborate on other contributing factors to the overall distance of the throw. Similar studies using better motion capture technologies and/or biomechanical modeling were also conducted after Dapena's study, with limited new perspectives [6,7]. Even parameters such as the earth's rotation were also identified as a contributor, somehow influencing overall distance [8]. In short, there has been very little, if any, 3D full-body hammer throw analysis to characterize the motor control of the hammer thrower. a holistic full-body control of the throw is still unclear. On the other hand, kinetic studies concentrated mainly on wire-tension characterization [2,9]. Other than the lack of the reliable 3D full-body kinematic data, the separation of kinematic and kinetic studies cannot supply a nexus of kinematic and kinetic data that reveals the control mechanism in order to improve hammer throw training.

Therefore, current training methods are largely based on coaches' subjective experiences of "what works". While this can be effective for some athletes, large and widespread biological diversity unfortunately limits the generalizability of individuals' experiences. Even small variations in limb length and muscle power generation, for example, can disrupt this form of knowledge transfer. Thus, the aim of this study is (1) to perform a holistic analysis in order to identify key parameters influencing throw quality, and (2) to establish a practical digital-route that can measure and quantify characteristics of an effective throw in order to supply science-based, real-time feedback for coaches and athletes to improve their training.

## 2. Materials and Methods

The hammer throw is an explosive sport [10]. In explosive performance, both force and velocity increase concurrently, therefore force measurement is often applied in quantification of explosive sports [2,11]. Wearable wire-tension measurement has garnered great interest in biofeedback training of the hammer throw [2,9]. They supply real-time, in-field/non-lab-based monitoring of tension/force generation as indicators of a trainee's performance progress. It seems that the hammer throw could be numerically analyzed in practice and the details of the motor control could be immediately available for coaches. However, the absence of a reliable method of linking the wire-tension data to the motor control of the throw has greatly hindered its application in practice. In order to bridge the gap, a synchronized measurement of 3D motion capture (kinematics of the throw/motor control) and wire-tension (kinetics of the throw) was applied to find the missing piece that could link the two types of data. One national-level athlete (body weight: 115 kg, body height: 178 cm, personal performance: 66.7 m) and one college-level athlete (body weight: 111 kg, body height: 176 cm, personal performance: 49.5 m) was tested, analyzed, and compared in order to find the link. We placed no restrictions on the subjects before and during the trials in an effort to preserve their normal motor control style. The university human-subject committee scrutinized and approved the test as to meet the criteria of ethical conduct for research involving human subjects. The subjects were informed on the testing procedures and voluntarily participated in the data collection.

### 2.1. Three-Dimensional Motion Capture and Biomechnaical Modeling

A twelve-camera VICON motion capture system (Oxford Metrics Ltd., Oxford, England) was set up on fully extended tripods around an indoor hammer pit with a safety set in front of the cameras. Six cameras were placed in a row parallel with the safety net on each side of the hammer throw pit. Capture occurred at a rate of 200 frames/second. Calibration residuals were determined in accordance with VICON's guidelines and yielded an accuracy within 1 mm. After warm-up, the national-level

athlete performed five trials and the college-level athlete performed six trials. The trial (judged by the fastest release speed) of each subject was selected, analyzed, and compared.

Each of the subjects wore a black garment made of stretchable material, which covered the upper and lower body. Affixed to the garment were 39 reflective markers, each with a diameter of 9 mm. Markers on the upper body were placed on the acromion process, lateral epicondyle of the humerous, styloid process of the ulna and radius, third metacarpophalangeal joint, as well as the upper and lower arm (again, no specific anatomical position is needed for these four markers), sternal notch, xiphoid process, C7, T10, and left back. Four markers were also placed on the head—one on the left and right temples each and two on the posterior portion of the parietal bone. Markers on the lower body were placed on the anterior superior iliac crest, posterior superior iliac crest, lateral condyle of the tibia, lateral malleolous of the fibula, calcaneal tuberosity, and the head of hallicus, as well as on the upper and lower leg (the four markers on the upper and lower leg were only used to determine segmental rotations. As they were not involved in segmental translations, no specific anatomical position was needed). Raw kinematic data was processed using a five-point (1-3-5-3-1 function) smoothing filter. From these 39 markers, a full body biomechanical model with 15 segments was built to reveal undisclosed aspects of the motor control [12–18]. The model worked as follows: from motion capture, we could establish anatomical positions, which then allowed the construction of a 15-segment full-body model. Using the fundamental precepts of physics, simple positional data were translated into the movement of the multi-segment model. In such individualized biomechanical modeling, the anthropometric characteristics of the body were established using anthropometric regression equations found in statistical studies [19,20]. The 15 segments were the head and neck, upper trunk, lower trunk, two upper arms, two lower arms, two hands, two thighs, two shanks, and each foot. In addition, three markers were attached on the handle. Furthermore, reflective tape was glued to the shot to determine hammer release speed. The following figure shows the set-up of the cameras, a sample frame of the 3D motion capture (Figure 1a), and the full-body model of the hammer throw (Figure 1b).

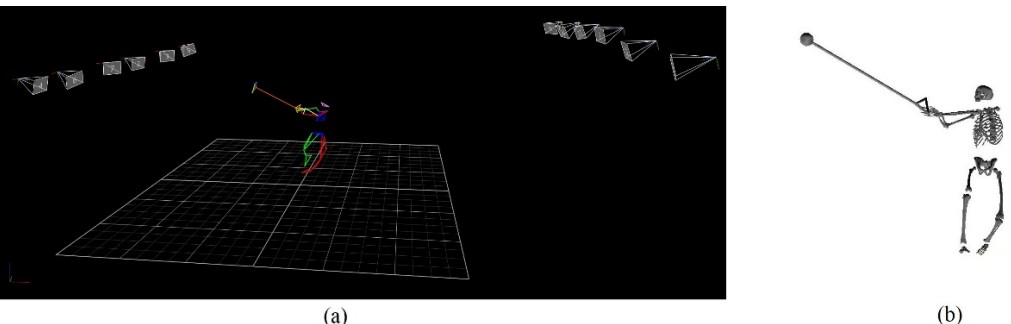

(a) (b)

**Figure 1.** (**a**) The set-up of the cameras and a sample frame of the 3D motion capture of the hammer throw; (**b**) The full-body biomechanical model of the hammer throw.

## 2.2. Wearable Wire-tension Measurement Unit

A wearable tension measurement unit [2] was used to capture the wire-tension variation of a throw. The data were used to reveal the throw force development, i.e., the characteristics of force generation during throwing. By comparing the elite athlete (national-level) thrower to the advanced (college-level) athlete, the effectiveness/force-generation during a throw could be quantified and revealed (note: this is not the motor-control of the body).

## 2.3. Motor Control Analysis for Feedback Training

Currently, there are a number of classification schemes used to describe the phases of a hammer throw, some focused on turning control [6,21,22] and others emphasized on the instances during hammer throw, e.g., (1) start of entry, (2) single support phase, (3) double support phase, and (4)

release [23]. These studies provide some perspectives of motor control on the hammer throw, and their collective shortcoming is that they fail to reveal the holistic control process. One especially important perspective—the relationship between force generation (kinetics) and the full-body control (kinematics) during throwing—is unclearly addressed in existing research, and as such, an understanding of the motor control remains incomplete for the hammer throw.

A simultaneous application of force/tension measurement and biomechanical modeling permit quantification of both kinetics and kinematics of human motor skills [24–26]. It is the only accurate way that can be used to make the connection between the throw-force generation and the motor control during a throw. As such, our approach provides the means to correlate the kinetic characteristics to kinematic ones. Furthermore, data comparison of different-level athletes is often applied in revealing relevant factors in motor learning [27–29]. Therefore, by comparing and abstracting the alterations between elite and advanced athletes, our study would be able to select key parameters, simplify the skill analysis, and establish a way of developing a new biofeedback tool on the motor-control optimization of the hammer throw.

## 3. Results

While a number of classification schemes have been used to describe the phases of the hammer throw, our results show that the skill should be divided into five phases: (1) initiation, (2) transition, (3) turns (normally three), (4) throw, and (5) follow through (Figure 2). The phases are defined by initial posture and trunk rotation, i.e., phase 1: from the beginning to the start of body rotation, phase 2: the 1st 360° trunk twist, phase 3: the 2nd–4th 360° trunk twists, phase 4: from the end of phase 3 to the release of hammer, and phase 5: from the release of the hammer to the end of the performance. The goal of initiation is to set the hammer into rotation and create the proper conditions to perform the transition. Our data shows that initiation is characterized by stationary-leg control as well as dynamic-arm control with two arm swings (i.e., forward and backward) and two overhead arm rotations (Figures 2 and 3). The transition is a period in which the control of the trunk and lower limbs shift from the stationary-leg control (i.e., upright standing) to the dynamic body turns. Our synchronized data unveils that the double-feet support (DFS) is much longer than single-foot support (SFS) during the transition phase, and the wire-tension displays the first big jump (Figure 2). The three turns demonstrate a gradual increase of the wire-tension's peak- and trough-value, with relatively equal DFS and SFS. The DFS aids the uphill development of the tension, while the SFS creates the downhill tension. The 3D motion analysis further reveals that the rotary plane of the hammer tilts gradually up, preparing for an optimal release (Figure 3). Following the turns is the throwing phase. The throwing phase is characterized by DFS and the second big jump of the wire-tension (Figure 2). The last phase is the follow through, which is used to dissipate the residual body angular momentum.

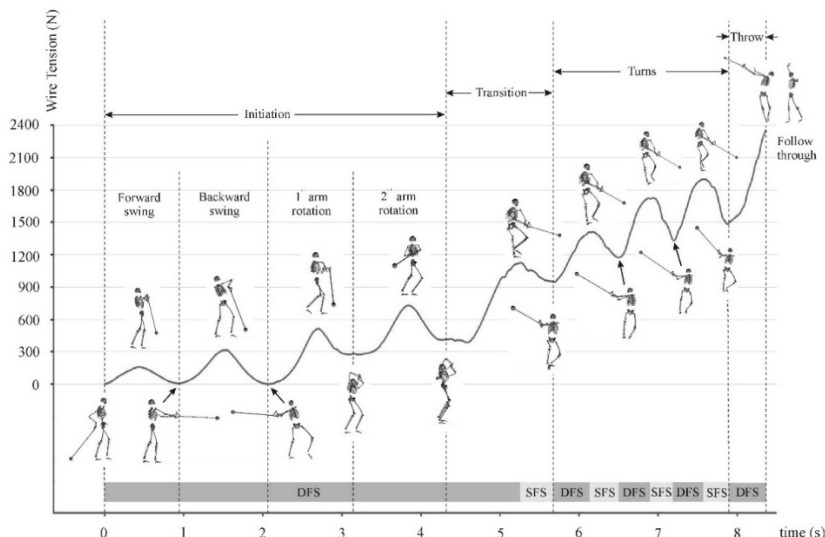

**Figure 2.** The development of wire-tension during a throw and the phase division based on both wire-tension variation and 3D motion analysis. The 3D biomechanical models reveal the postures at key instances (dotted lines). DFS—double feet support, SFS—single foot support.

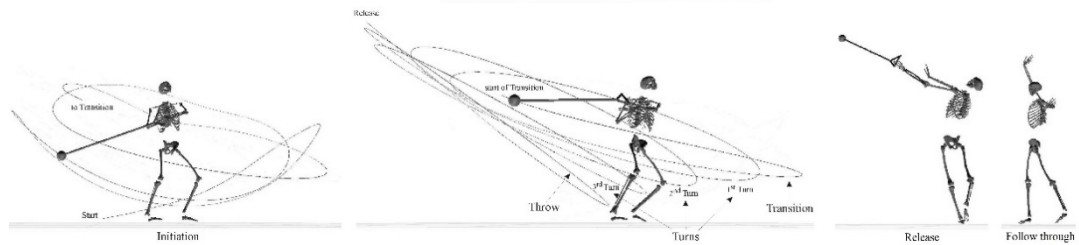

**Figure 3.** The spatial trace of a hammer in various phases and some dynamic postures on key time points.

Based on numerous analyses of our synchronized data, the following variables have been selected for illustrating the characteristics of the quick and complex throw skills in order to establish a practical, digital way to optimize the practice:

- timing
- the tension development/force-generation
- peak-to-trough values during turns
- upper-limb control/vertical variation of hands
- trunk control/trunk tilt angle
- lower-limb control/vertical variation of hip.

Such focused communication would help practitioners understanding the complex motor control in a timely, efficient way.

Table 1 represents the timing characteristics of the hammer throw. In general, the national-level subject could finish the throw faster than the college-level subject. Our data illustrates that, compared to the latter, the elite thrower took more time for initiating the rotatory movement of the hammer (i.e., slower initiation), but could perform a faster transition, quicker body spins, and more explosive throws (i.e., shorter throw phase). Further, the timing data indicates that the movement control of the elite thrower became consistently faster, while the college-level subject was unable to keep such a trend.

**Table 1.** The timing characteristics and their difference between the national-level and the college-level athlete.

| Duration (s) | National-Level | College-Level | Difference |
|---|---|---|---|
| Initiation | 4.66 | 4.24 | 9.9% |
| Transition | 1.09 | 1.38 | −21.0% |
| 1st Turn | 0.77 | 0.82 | −6.1% |
| 2nd Turn | 0.54 | 0.68 | −20.6% |
| 3rd Turn | 0.46 | 0.70 | −34.3% |
| Throw | 0.41 | 0.54 | −24.1% |

Tables 2 and 3 show the kinetic characteristics of the hammer throw. The tension developments of both subjects evidence two different strategies of throw-force generation. The elite thrower demonstrated three explosive-force generations (i.e., the transition, the third turn and the throw), while the college-level thrower had only two (i.e., the transition and the throw). Compared to the latter, the former had lower tension in the initiation, the first turn, and second turn, but was able to surpass the latter in the third turn and in the throw by 12.7% and 21.4%, respectively. Furthermore, the P-T (peak-to-trough) values reveal that the former was able to control the tension in a larger range than the latter, indicating a better dynamic control of the former.

**Table 2.** The characteristics of force-generation and their difference between the national-level and the college-level athlete.

| Tension (N) | National-Level | College-Level | Difference |
|---|---|---|---|
| Initiation | 371 | 411 | −9.8% |
| Transition | 975 | 947 | 3.0% |
| 1st Turn | 1023 | 1172 | −12.7% |
| 2nd Turn | 1184 | 1330 | −11.0% |
| 3rd Turn | 1673 | 1484 | 12.7% |
| Throw | 2853 | 2350 | 21.4% |

**Table 3.** The characteristics of the tension control and their difference between the national-level and the college-level athlete.

| P-T Value (N) | National-Level | College-Level | Difference |
|---|---|---|---|
| Transition | 205 | 178 | 15.1% |
| 1st Turn | 587 | 238 | 146.7% |
| 2nd Turn | 704 | 395 | 78.3% |
| 3rd Turn | 463 | 416 | 11.3% |

P-T value: peak-to-trough value in each phase or turning cycle.

Tables 4–6 are related to the kinematics/segmental control of the hammer throw [3]. Consistent with the kinetic characteristics, the elite thrower showed more dynamic control of the upper limb during turns than the college-level thrower. As for the trunk control, our data demonstrated that the trunk was dynamically involved in the initiation and transition (i.e., large P-T values, Table 5) and decreased the tilting (i.e., small P-T values) during the turns for both subjects, but only contributed to the throw for the elite thrower only (substantial re-increase of the P-T value). The remarkable difference between the subjects shows that the elite thrower demonstrated stable control, i.e., a gradual increase in the movement range of both arms and legs from the first turn to the third turn, while such a control was absent for the college-level athlete. Additionally, despite close body-heights (i.e., 1.78 m vs. 1.76 m) and release heights (1.45 m vs. 1.44 m), the elite athlete had a higher hip height at release than the college-level athlete. These kinematic data indicates that the elite thrower has mastered an effective leg control and/or a good coordination between the upper and lower limbs.

**Table 4.** The characteristics of the upper-limb control and their difference between the national-level and the college-level athlete.

| Hand Vertical Height (m) | National-Level | College-Level | Difference |
|---|---|---|---|
| P-T value of Transition | 0.78 | 0.85 | −9.0% |
| P-T value of 1st Turn | 0.59 | 0.41 | 30.5% |
| P-T value of 2nd Turn | 0.66 | 0.44 | 33.3% |
| P-T value of 3rd Turn | 0.72 | 0.62 | 13.9% |
| Release | 1.45 | 1.44 | 0.7% |

P-T value: peak-to-trough value in each phase or turning cycle.

**Table 5.** The characteristics of trunk control and their difference between the national-level and the college-level athlete.

| P-T Value of Trunk Tilt (°) | National-Level | College-Level | Difference |
|---|---|---|---|
| Initiation | 33.6 | 35.7 | −6.2% |
| Transition | 29.9 | 38.6 | −29.1% |
| 1st Turn | 6.3 | 18.8 | −198.4% |
| 2nd Turn | 11.8 | 12.6 | −6.8% |
| 3rd Turn | 14.4 | 14 | 2.8% |
| Throw | 29.7 | 16.8 | 43.4% |

P-T value: peak-to-trough value in each phase or turning cycle. Trunk tilt: the orientation of the trunk, i.e., the angle between trunk and horizontal plane.

**Table 6.** The characteristics of the lower-limb control and their difference between the national-level and the college-level athlete.

| Hip Vertical Height (m) | National-Level | College-Level | Difference |
|---|---|---|---|
| P-T value of Transition | 0.12 | 0.14 | −16.7% |
| P-T value of 1st Turn | 0.09 | 0.07 | 22.2% |
| P-T value of 2nd Turn | 0.11 | 0.15 | −36.4% |
| P-T value of 3rd Turn | 0.12 | 0.09 | 25.0% |
| Release | 1.11 | 1.06 | 4.5% |

P-T value: peak-to-trough value in each phase or turning cycle.

## 4. Discussion

There are several new aspects revealed by the current study. All the aspects would be vital for training and optimizing the hammer throw. Due to the limited studies on hammer throw, there is currently no review article available to summarize the key factors influencing the quality of the throw. An equivalent one is a book chapter published in 2000 [30]. Hence, these new aspects would also play a relevant role in developing digital method in coaching the throw skill.

Firstly, the initiation phase is totally overlooked by researchers. Our data indicate that elite thrower completes 13.0% (i.e., 371/2853 in Table 2) of maximum force in the initiation, while the college-level athlete finishes 17.5% (i.e., 411/2350 in Table 2). Generating more percentage force, i.e., setting the hammer rotation faster, during the initiation may affect the entrance to the next phase, i.e., setting the body into spin to follow the hammer. An unsuitable condition for entrance may delay the increase of the tension, e.g., the tension plateau of college-level athlete in Figure 4. a right pace of the initiation should be individualized based on the physical ability and training experience. a real-time biofeedback device may help the individualized optimization of the initiation phase [4].

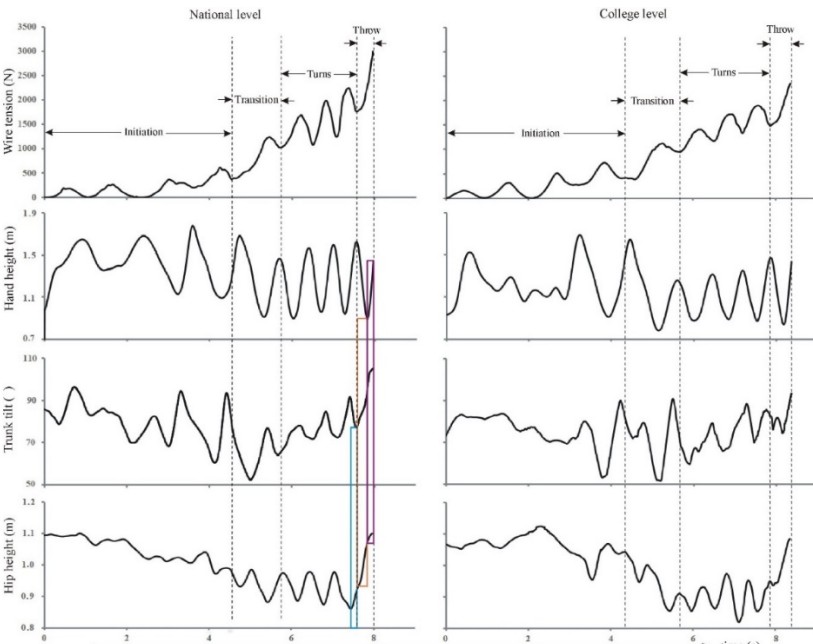

**Figure 4.** Throw-force developments and their possible control patterns of hammer throw revealed by synchronized data of wire-tension and coordination between upper and lower limb.

Secondly, the transition phase is inappropriately defined in the previous literature. The widely accepted definition for hammer throw analysis is the four rotation-release model [6,30]. However, both our kinetic/tension and kinematic/3D data reveal that the first body spin is essentially different from the rest of the body spins (Figure 4). Both athletes perform their first explosive-force generation, reaching 34.2% of max tension for the national-level athlete (i.e., 975/2853 in Table 2) and 40.3% of max tension for the college-level athlete (i.e., 947/2350 in Table 2). Such a rapid increase is not repeated in the following body turns. Additionally, the first turn has ca. three times longer DFS than SFS, while the rest turns have ca. equal DFS and SFS. Further, the first turn illustrates notably different leg control (i.e., the up-and-downs of hip movement) when compared to the rest of the turns. Even more dramatic, the controls are different between the two athletes. Obviously, more subjects are needed for future studies in order to reveal if there exists general control-patterns. Therefore, the first body rotation could function as a connector between the stationary-leg control and the dynamic full-body turns, i.e., a transition phase between the initiation and the turns. It is an adaptation from one motor control pattern to the other [31,32]. Such an identification is conceptually important in training the switch between standing and dynamic posture control. It is also practically relevant in the evaluation of skill efficiency and biofeedback effectiveness.

Thirdly, our data establishes a practical way that could easily identify the effectiveness of a throw. The hammer throw is a full-body movement, requiring multi-joint coordination. Three-dimensional motion capture is currently the reliable method for demystifying the control patterns of multi-joint coordination in human full-body movement [33–35]. However, practitioners find that the 3D motion capture technology is too cumbersome to be useful in practice [2,36]. Through our synchronized data analysis, we have found that hand and hip vertical height as well as the trunk tilt could be used to reveal the multi-joint coordination. When considering the effectiveness of the full-body coordination in generating explosive force, it is important to note that the whip-like control mechanism should be followed [13,14], i.e., the generation of the explosive-force should first start in strong/big muscle groups and follow the big-to-small-muscle sequence flow. In case of throwing the hammer (i.e., the throw phase), a performer should firstly extend the legs, then the trunk (i.e., trunk extension), and lastly perform the shoulders/arms throw for the release of the implement. Due to the force required, the explosive control (as fast as possible) and the multi-joint coordination, such a whip-like

performance in a hammer throw needs, obviously, years of strength and coordination training in order to improve its effectiveness. Figures 4 and 5 show that the elite athlete had formed the whip-like control, i.e., leg extension first (highlighted by a blue rectangle in Figure 4), followed by trunk extension with further leg extension (highlighted by an orange rectangle in Figure 4), and explosive arm up-swing with the final leg and trunk extension (highlighted by a violet rectangle in Figure 4). Such a whip-like throw was not found in the performance of the college-level athlete. The short plateau in the leg extension and up-and-downs in the trunk extension (Figure 4) as well as not-fully-extended leg and trunk may suggest a weak leg and trunk and/or poor leg-trunk coordination of the college-level athlete. Additionally, SSC (stretch-shortening cycle) is the other mechanism, influencing the generation of muscle tension/force [37]. a properly dynamic pre-lengthening of the muscle groups involved would improve the effectiveness of the throw-power generation. Our data (Figure 4) indicates that the elite subject could apply the SSC principle in all the turns and the throw in a more efficient way than the college-level subject.

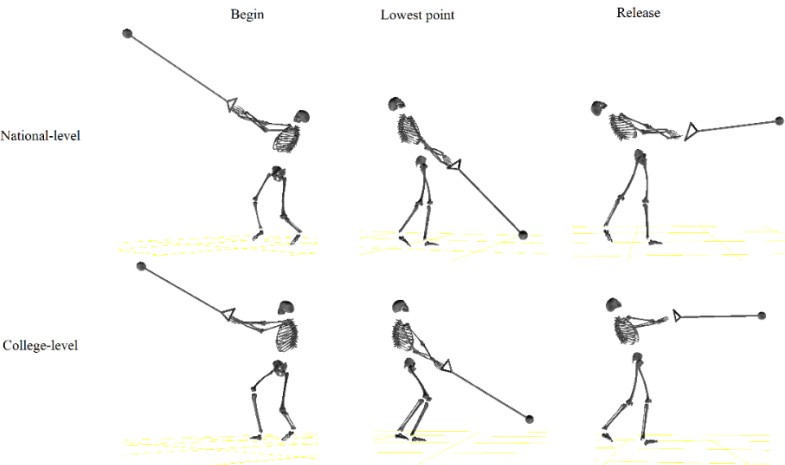

**Figure 5.** The kinematic comparison of the throw phases and dynamic postures between the national- and college-level athlete.

It is well known that every study has its limitations. The two possible limitations of the current study could be: (1) the selection of the key parameters was based on the comparison of two athletes, and (2) the performance quality was defined by using wire-tension. However, these types of limitations are the unavoidable trade-offs in any technological development, i.e., one has to start a scientific quantification with some degrees of simplification; and we are aware that these trade-offs/simplifications can be remedied by future studies. In the case of the development of biomechanical feedback training in the hammer throw, our current study may make the initial move. For overcoming the first limitation, more elite and advanced athletes should be tested, analyzed, and statistically compared to confirm if there are more key parameters contributing to the throw distance. If yes, investigations should be launched, focusing on adding more wearable sensors to the current development. As for the second limitation, previous studies have shown that hammer release angle is the secondary factor [38]. However, wearable sensors for detecting hammer-release angle need to be developed first, and then a new system could be developed to add the tracking of the hammer-release angle to the biofeedback devices.

In summary, our study demonstrates that the speedy and complex multi-joint control of the hammer throw might be quantified and characterized by four key parameters: wire-tension, hand and hip height, and trunk tilt. As such, four sensors (i.e., one tension sensor and three IMU (Inertial Measurement Unit [39,40]) sensors) would be sufficient to build a digital system for realizing a real-time biofeedback system in practice. The encouraging reality is that both types of sensors can be applied in wearable devices [2,3,39]. This means that an easily applied wearable system could substitute the lab-based,

synchronized measurement of tension and 3D motion capture technology. Ergo, our study would supply a methodological breakthrough in digital-aided sport training. It would allow practitioners to develop wearable-based, real-time biomechanical feedback devices that could transform the hammer throw learning and training paradigm from a largely subjective art into a precise scientific method. The results of our study have great potential to: (1) transfer the scientific monitoring of the hammer throw from a lab-based environment into the field, (2) simplify a scientific movement quantification, transitioning from using a complicated and time-consuming 3D motion capture system to easily-applied wearables, and (3) transfer the vital biomechanical feedback in real-time to prevent movement errors when learning, while finding individual compensation and/or optimization to improve throw effectiveness.

## 5. Conclusions

The quick and complex hammer throw can be digitally quantified by simultaneously monitoring the wire-tension, the vertical movements of hands and hips, as well as the trunk tilt. These four parameters might reveal the relationship between tension/force generation and the full-body coordination. The throw force should be gradually developed and materialize in the following four phases: the initiation, the transition, the turns (normally three), and the throw. Among them, the transition, the third turn, and the throw should produce explosive/rapid increases in tension. The effectiveness of the throw-force generation, whip-like control of the full-body, and proper SSC of muscle groups involved should be established through years of training. a wearable digital device with tension and IMU sensors would have great potential to realize a real-time biomechanical feedback training in practice for evaluating and improving the efficiency of various training programs.

**Author Contributions:** B.W., Y.G., X.Z. and G.S. designed the study. B.W., Y.G., X.Z. and Y.W. performed the data collection. Y.W. and H.L. provided the technical support. B.W., Y.G., X.Z. and G.S. analyzed and interpreted the data; B.W., Y.G. and G.S. prepared the draft; all authors contributed to the revisions and proof reading of the article. All authors have read and agreed to the published version of the manuscript.

**Funding:** The research project was supported by National Sciences and Engineering Research Council of Canada (NSERC).

**Conflicts of Interest:** The authors declare no conflict of interest. The founding sponsors had no role in the collection, analyses, or interpretation of data; in the writing of the manuscript, and in the decision to publish the results.

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
