# Peer review of "Hammer Throw: A Pilot Study for a Novel Digital-Route for Diagnosing and Improving Its Throw Quality"

_applsci, doi:10.3390/app10061922_

Round 1

Reviewer 1 Report

Review Report – ApplSci - 691269

• A brief summary  

In this study authors presented biomechanical approach of analysis of key parameters influencing hammer throw quality, dominantly trough kinematics and kinetics, especially with regard to the fact that the world record of hammer throw has not been broken since 1986.

• Broad comments  

Introducing overview was supported by methodology used to fulfil aim of the experiment. Experimental design was generally appropriate with adequately presented methods, followed and supported by relevant and precise conclusions. Limitations of the study were not presented adequately, especially with regard to best throw selection criteria and performance quality level differentiation criteria. However, it is an upgrade of previous research with significant scientific and practical contributions.

• Specific comments

Ln 45-6: “The best trial (judged by the fastest release speed) of each subject was selected, analyzed and compared.” - It is a mission to produce fastest release speed at best throwing hight and angle. Since it is only one criteria (among more) for trial to be successful, it may be wrong to comment and argument ‘dynamic control’ later in text (e.g. page 6, line 15). Suggestion would be to take word ‘best’ out of criteria definition, or to change interpretations about ‘dynamic control’ later. please define starts and endings of phases more precisely, especially with regard to ‘initiation’ and ‘transition’ Bring limitations of the experimental design more boldly within one or two sentences

Reviewer 2 Report

This paper discusses the combined usage of kinematic and kinetic measurements in the quantitative analysis of hammer throw sport, with the goal of providing insight and technological tools for couching. 

The paper is well-written and easy to read. The research question (how to improve hammer throw couching) is valid and of interest to the related audience. The presented methodology seems to be an improvement to the existing approaches to quantify kinematics and kinetics of the movement in this sport. 

My major comment is that the data (analyzing and comparing only two throws) is very limited and cannot be used to validate any of the claims. Playing devil's advocate (understandably I am probably viewed by the authors as the devil himself!!), I argue that the shorter initiation phase of the elite athlete is detrimental to his performance. To prove that I am wrong, statistical analysis is needed. There is a good chance that their findings are, in fact, true. However, with such limited data, it is quite likely that the results are not valid. 

One possible way to improve the results is, at the very least, to consider all the throws by the athletes and then compare the average. This way, you can tell whether there is "a consistent pattern" for the elite athlete, which outperforms the college-level one. 

I understand that performing experimental trials, especially with national-level athletes, is a big logistic headache. But I think it is necessary in order to claim that there is "a superior" pattern. Maybe what you have observed is just this athlete's style, and others may perform equally well with a completely different style. The claim that you have identified the winning pattern cannot be supported by your current data.

My other comments are as follows.

Power vs tension vs energy. Reading through the manuscript, I noticed a rather haphazard mixture of these concepts, and sentences/claims need to be revised to be more scientifically accurate. I get this impression that the authors correlate tension with power (e.g. saying percentage power when discussing the ratio of tensions). These quantities are somewhat related, but not in a very direct way. A thorough revision of the technical language is required. In a related line of thoughts, what happens to the energy of the ball? My understanding is that, at the end of the day, all that matters is the velocity of the hammer (and its angle) that contributes to how far it will travel. The velocity (equivalently the kinetic energy of the ball) is more informative (and interesting to me) than the wire tension. You can simply calculate 0.5*m*v^2, and plot that along with the rest of the variables. Having that, then you can say when the energy of the hammer goes up, you have power generation (better yet, transferring power to the hammer). I noticed many unnecessary usages of the hyphen (-) in the text. E.g. wire tension, upper-limb, lower-limb, peak-value, posture-control, big-jump, etc. (and they are used inconsistently). please revise the text. I also found the term "digital" used a bit outdatedly (personal opinion). The way it is used reminds me of the early 90s where things were transitioning from analog to digital. Now everything, including the report the couches write, is "digital"! I feel like terms such as, "quantitative system", or "computer-assisted" serve the purpose better. Food for thoughts.  To people not in the hammer throw field, the notion of wire-tension measurement is not as obvious as it may sound. I didn't know what the authors are referring to until section 3.2 that I realized that it simply means measuring the tension in the wire in the hammer. I suppose more context in the introduction can be helpful for the out-of-field audience.  Again related to terminology, the term quasi-static is not used correctly. Quasi-static means that changes in the system are so small that you can assume they are constant. In this context, although the legs don't move much, they transmit large (and varying) forces, and their control by no mean is quasi-static. A better term to use here is perhaps "stationary legs".  How were the phases identified? What kinematic/kinetic variable and what threshold was used to define the transition from one phase to another? Was it eyeballing? If so, what are the implications of errors in locating the transition points? This needs to be addressed and discussed in the manuscript. Some terms were not properly defined before they were used for the first time. E.g, P-T (line 31 page 5). Also on lines 31 and 32 on page 7, there is "ca.", which does not seem to belong here.  The results presented in tables 4 and 6 compare the absolute values. It is likely that the athletes' height and intersegmental proportions contribute to some of the differences. These values are better to be normalized (e.g. to height, or arm/leg length). 

Round 2

Reviewer 2 Report

I would like to thank the authors for their responses.

Unfortunately, I am not fully satisfied with the limited revisions in the updated manuscript.

My major concerns are unfortunately unmet, and my comments remain the same for the revised manuscript. 

Author Response

Dear Reviewer,

Thank you for your comments.

We would appreciate very much if you could provide specific feedbacks. The current comments would be too general to be helpful for a scientific debate and convince. In our previous responses, we have addressed all of your comments point-by-point; some suggestions are well taken and revisions have been made, while others are respectfully disagreed with our rational. We would be grateful if you could let us know which of our responses have satisfied you and which remain unmet to you? For the unmet ones, could you please provide your rational? In our opinion, a rationalized debate would be able to identify the necessary elements that should be included in writing a clear (i.e. logically constructed) and concise (i.e. stay-on-topic) article.

Again, thank you for your time and expertise!

Prof. Dr. Gongbing Shan

Round 3

Reviewer 2 Report

Dear Authors,

In my opinion, this manuscript has definite merits and the results are novel. I wish to see the results published. However, as I had pointed out in my first correspondence, there are a few major issues that need to be addressed to make the paper publishable.

Firstly, the general message of the paper. The manuscript in its current form almost entirely focuses on presenting the new results. I am not convinced by the authors' response that "the focus of this paper is technology development" as the paper is heavily results-centered. I understand that these results are meant to provide the basic rationale for developing a new training system. Unfortunately, however, this paper lacks the rigor to prove the point that the observed patterns are indeed "the" pattern. Therefore, I cannot easily accept the final conclusion that the four suggested key parameters are the right set of measurements to target. In fact, my hunch is that the "style" varies quite substantially among even national-level athletes, which may dismiss your rationale.

I understand that bringing national-level athletes to the lab is a huge logistic problem. But I had suggested a cheap and available alternative. At least, instead of analyzing only the best trial, take the average of all or the best three trials. This way, at the least, you can take within-athlete variability into account. Also, may I disagree with you that one trial is by no means "plentiful synchronized data" (line 173).

My other important comment was on the relationship between force (tension), energy and power. For instance, I wonder why the authors have preferred to use the term "power generation" instead of "tension build-up" when discussing the increase in tension. The tension and energy are related, but the relationship is rather complex as the energy transfers back and forth between the body and the hammer. (Side note, a better justification would have been that power is related to the rate of change of tension. p=F.v is actually not relevant here at all). I recommend the authors use the scientifically correct terminology (use "tension" when discussing tension, "power" when discussing power, "energy" when discussing energy). I also recommend the put a note in the paper explaining what the relationship between these variables is.

May I also suggest looking into the velocity of the hammer. Please view this as a suggestion made by a colleague who wishes to help to improve your technology. I am not asking you to do it in this paper, but you may think about it in developing your technology. Wire tension by itself is not the objective of this sport. It is a proxy measurement for the energy of the hammer (and ultimately its velocity vector, which is "the" objective). My guess is that making an instrumented hammer (with an IMU inside) gives you the same (or better) information. Also, building a measurement system that is made entirely with IMUs is probably easier than including a wire-tension system. 

Round 4

Reviewer 2 Report

Dear authors,

Thank you for crafting this response. My opinion is that one data point is not enough for this paper. You are happy with one. Clearly we cannot agree on this. 

I also thank you for replacing all the power and energy words with force. It is better now. Although, it was not what I meant. I think you were better off on some occasions to talk about power or energy (e.g. line 39, explosive power generation was perfectly fine). 

This is my last round of comments. I will leave it to the editors to decide.

Best of luck.

Author Response

Dear reviewer,

The last suggested change (i.e. “line 39, explosive power generation was perfectly fine”) has been made.

Thank you again for your time and expertise!

Gongbing Shan